# Highly stretchable polymer semiconductor thin films with multi-modal energy dissipation and high relative stretchability

Hung-Chin Wu [1], Shayla Nikzad[1], Chenxin Zhu[2], Hongping Yan[1,3], Yang Li [4], Weijun Niu[4], James R. Matthews[4], Jie Xu [1,6], Naoji Matsuhisa[1,7], Prajwal Kammardi Arunachala[5], Reza Rastak [5], Christian Linder[5], Yu-Qing Zheng[1], Michael F. Toney [3,8], Mingqian He[4] ✉ & Zhenan Bao [1] ✉

Stretchable polymer semiconductors (PSCs) have seen great advancements alongside the development of soft electronics. But it remains a challenge to simultaneously achieve high charge carrier mobility and stretchability. Herein, we report the finding that stretchable PSC thin films (<100-nm-thick) with high stretchability tend to exhibit multi-modal energy dissipation mechanisms and have a large relative stretchability (rS) defined by the ratio of the entropic energy dissipation to the enthalpic energy dissipation under strain. They effectively recovered the original molecular ordering, as well as electrical performance, after strain was released. The highest rS value with a model polymer (P4) exhibited an average charge carrier mobility of 0.2 cm²V⁻¹s⁻¹ under 100% biaxial strain, while PSCs with low rS values showed irreversible morphology changes and rapid degradation of electrical performance under strain. These results suggest rS can be used as a parameter to compare the reliability and reversibility of stretchable PSC thin films.

Polymer semiconductors (PSCs) have been developed for use in next-generation soft electronics (e.g., flexible, bendable, or stretchable devices)[1,2]. Both chemical design strategies and fabrication methods have been explored to enhance their electrical and mechanical performance. PSC thin films can now realize comparable and even superior electronic performance to that of amorphous silicon[3]. However, to truly harness the advantages of flexibility and stretchability of polymers, it is necessary to develop PSC thin films with good stretchability that maintain electrical performance under repeated cycles of strain. Such semiconducting thin films are essential in order to create mechanically robust electronics[4–13].

Chemical and physical approaches, such as tuning rigidity of the polymer backbone[14–19], inter-chain crosslinking[20,21], incorporating energy dissipating dynamic crosslinking[22], adding molecular additives[23–25], polymer blending[26–29], nanoconfinement[30,31], side-chain modification[18,20,32], block-copolymerization with flexible polymers[33,34], and tailoring polymer molecular weights[35–37] are reported as approaches for producing good mechanical properties under external strains. However, few of these approaches have achieved both high stretchability and good electrical properties under strain. In general, PSC thin films, typically less than 100 nm in thickness for field-effect transistors (FETs), possess various levels of conformational freedom from

[1]Department of Chemical Engineering, Stanford University, Stanford, CA 94305, US. [2]Department of Electrical Engineering, Stanford University, Stanford, CA 94305, US. [3]Stanford Synchrotron Radiation Lightsource, SLAC National Accelerator Laboratory, Menlo Park, CA 94025, US. [4]Corning Incorporated, Corning, NY 14831, US. [5]Department of Civil and Environmental Engineering, Stanford University, Stanford, CA 94305, US. [6]Present address: Nanoscience and Technology Division, Argonne National Laboratory, Lemont, IL 60439, US. [7]Present address: Institute of Industrial Science, The University of Tokyo, Meguro, Tokyo 153-8505, Japan. [8]Present address: Department of Chemical and Biological Engineering and Renewable and Sustainable Energy Institute (RASEI), University of Colorado Boulder, Boulder, CO 80309, US. ✉e-mail: hem@corning.com; zbao@stanford.edu

polymer backbones and side chains, which can result in complex processing-dependent morphologies with both crystalline and amorphous domains coexisting[1,5]. Both the ability to transport charges and to dissipate mechanical stress in semicrystalline polymer thin films depend on molecular conformation, packing structure, grain size, degree of crystallinity and connectivity between grains[5]. Under an external strain, the larger crystalline domains may fracture and reduce electrical performance, or the thin film may yield under strain and fail to relax back to its original dimensions. None of the above issues are desirable for practical robust soft electronics[1–5].

A number of properties, such as elastic modulus, tensile strength, resilience, and fatigue life, are standard parameters used to characterize polymer bulk materials and thin films. However, there is a need for parameters to directly compare polymer semiconductor thin films used specifically for thin film transistor applications, which have thicknesses less than 100 nm and typically are supported by elastic substrates. Some of the above-mentioned parameters may be challenging to measure for ultrathin films. As a result, crack onset strain is among the most commonly used, due to the simplicity in measurement, to compare stretchability of polymer semiconductor thin films[5,14]. However, it is highly dependent on molecular and microstructural changes which impact changes in mechanical and electrical behavior of polymer semiconductors under repeated strain cycles. The morphology of PSC thin films is well-known to be dependent on processing conditions, film thickness, and molecular weight, in addition to the polymer chemical structure. However, crack onset strain does not always correlate well with observed degradation of polymer semiconductor thin film electrical performance behavior under strain.

Since molecular conformation and microcrystalline morphology changes are important mechanisms for energy dissipation under strain, we propose a new parameter, relative stretchability ($rS$) to compare the relative strain tolerance of various PSC thin films for transistor applications. $rS$ is defined as the change of polymer chain alignment (i.e., dichroic ratio (DR)) over the change of relative degree of crystallinity (rDoC) as a function of strain, $rS = \Delta DR / \Delta rDoC$. The degree of chain alignment under strain directly correlates with energy dissipated entropically through polymer conformation change and crystallite re-orientation assuming similar viscoelastic properties; while the change of rDoC ($\Delta rDoC$) represents the amount of energy dissipated enthalpically through amorphization of crystallites assuming similar latent heat of fusion (taking into account that there may be crystal formation due to strain-induced crystallization). Therefore, $rS$ presents the ratio of the entropic energy dissipation over the enthalpic energy dissipation. Note that strain-induced crystallization is a more common phenomenon for semicrystalline polymers[25,38,39]. Strain-induced changes in crystallinity in polymer semiconductors only became of interest recently and has not been extensively investigated. However, we have observed such a phenomenon in several polymer semiconductors we investigated previously and in our current systems[5,16–25]. The observed overall change of rDoC on PSC thin films may be from a competition between strain-induced crystallization and strain-induced amorphization depending on the specific thin film crystalline domain fraction, size, and distribution as well as polymer backbone rigidity and level of local aggregation in the amorphous domains. Due to the high rigidity of conjugated structures, a higher stress is asserted on the crystalline regions during strain when compared to the more flexible non-conjugated polymers. In this report, diketopyrrolopyrrole (DPP)-based PSCs, P2TBDPP2TBFT4, with molecular weights from approximately 20–100 kg mol⁻¹ (denoted P1–P4; Fig. 1a)[40] are used as model compounds while data from reported polymers are used for additional comparisons. Our results show that $rS$ provides a good comparison of the likelihood of various PSC thin films to maintain stable thin film transistor electrical performance under repeated strain cycles regardless of their chemical structures. This finding also suggests that similar underlying molecular mechanisms

contribute to stretchability of PSC thin films used for FETs. A higher $rS$ value was found to correlate to a stronger ability for thin film deformability and retention of electrical functionality under strain, which arises from the ease of molecular conformational and chain alignment, while resisting fracturing in the crystalline domain fracturing. Indeed, P4, a high molecular weight PSC, exhibited the above molecular behavior and a high $rS$. Correspondingly, P4 maintained a high FET mobility of 0.2 cm²V⁻¹s⁻¹ under 100% *biaxial* strain (i.e., 400% area expansion) without noticeable degradation compared to pristine films. Surprisingly, we observed that despite its rigid conjugated polymer backbone, P4 displayed reversible chain alignment along arbitrary strain directions, which is rare for PSCs and more common for classical rubbery elastic polymers[41]. This work provides important fundamental insights for development of mechanically robust stretchable polymer thin film electronics.

## Results

### Multi-modal energy dissipation mechanisms and relative stretchability

Achieving high thin film deformability without compromising electrical performance has been a long-standing challenge for PSCs. Previous work focused mainly on the effects of uniaxial strains[1–5]. A systematic correlation between morphological, mechanical, and electrical behaviors based on molecular organization needs to be developed, especially for stretchable PSCs that in application are likely to endure multi-directional deformation (Fig. 1a). Here, we uncover the type of PSC thin film morphology that enables stable performance under biaxial strain cycles (Fig. 1c). P2TBDPP2TBFT4-based PSCs with number-averaged molecular weights of 22 (P1), 50 (P2), 73 (P3), and 97 (P4) kg mol⁻¹ are studied (Fig. 1a). Above a critical molecular weight, these polymers showed a clear increase in the slope of solution viscosity as a function of molecular weight, suggesting the presence of topological entanglements, chain aggregation, or change in solubility (Supplementary Fig. S1), while the polymer thin films became more disordered[37,42,43]. The glass transition temperatures ($T_g$) of the conjugated backbone were only visible for bulk P3 and P4 films (>1 μm in thickness, Supplementary Fig. S2). This implies that P3 and P4 thin films are likely more disordered. Note that the $T_g$ of sub-100-nm thin films (i.e., thin films for stretchable transistor devices), which usually requires specialized methods to measure, are known to differ from the bulk film properties[44]. Furthermore, the molecular packing orientations of the crystalline domains in P1–P4 films were characterized using grazing-incidence X-ray diffraction (GIXD) (Supplementary Fig. S3). P1 exhibited a strong edge-on packing preference, while P2–P4 had both edge-on and face-on packing orientations with the fraction of face-on crystallites significantly increasing with molecular weight. Furthermore, the charge transport behavior was affected by the different crystallite orientations (Table 1; Supplementary Fig. S4)[43,45].

In addition to packing orientation, the rDoC is an important parameter that impacts the electronic and mechanical characteristics of stretchable PSC films. As is well established for other reported PSCs[35–37], the rDoC usually drops progressively with increasing molecular weight (Supplementary Fig. S5), and the semicrystalline films became more disordered for higher molecular weight polymers. Comparing P1–P4, the rDoC value decreased by over 93%. Such a large change in crystallinity has been directly linked with thin film deformability. It was seen that the lower molecular weight P1 and P2 films were brittle and could easily form macro-cracks at less than 30% strain, while thin films (~50 nm) based on high molecular weight P4 can be stretched to over 100% strain without cracks (Supplementary Fig. S6 and 7). Consistent with the above observations, a P4 bulk film (approximately 1.5 to 2 μm) can be stretched to over 35% strain, while other PSC bulk films typically break below 20%[46] (Supplementary Fig. S8). Moreover, the elastic moduli of P1–P4 films were measured in a range from 20 to 350 MPa (Supplementary Fig. S9), while lower moduli were observed

for higher molecular weight films, again consistent with their much lower rDoC.

Although the above results show that molecular ordering and morphology of PSCs can directly impact thin film properties, none of these properties on their own can be used to determine whether a given PSC will exhibit stable and reversible electronic performance under strain[1,3,4,42]. Moreover, it is challenging to compare the promise of various reported intrinsically stretchable PSC thin films, typically less than 100 nm thick, for thin film transistors, owing to variations in characterization and sample preparation methods used. Therefore, a quantitative metric is needed to allow comparison of PSC's ability to maintain high electrical performance under

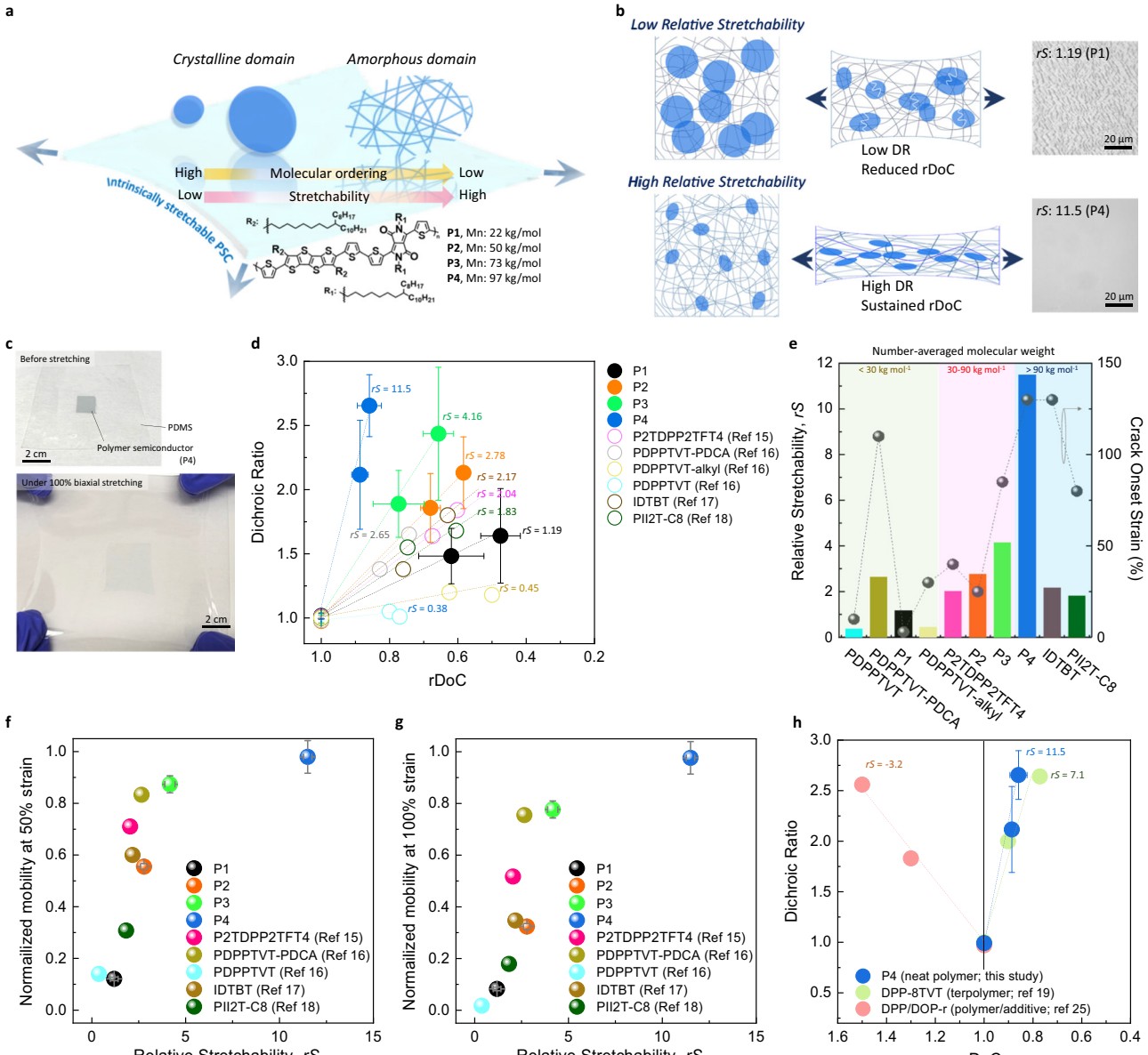

**Fig. 1 | Molecular ordering guided PSC properties. a** Schematic illustration of the impact of molecular ordering on thin film mechanical properties. Desirable molecular organization is required to allow an intrinsically stretchable PSC film to be stretched under not only uniaxial but also multi-directional strains. **b** Illustration of polymer chain dynamics under strain in a polymer film that possesses high or low stretchability. **c** A polymer film under multi-directional strain. The PSC film (i.e., P4) is prepared and stretched biaxially on a PDMS substrate. The stretching forces are applied uniformly from 4 directions, and a maximum strain of 100% is introduced biaxially. **d** Calculated relative stretchability (rS) values of reported neat stretchable PSCs are depicted and compared with P1–P4 in this work. rS values of P1–P4 were averaged from at least 2 DR and 2 rDoC samples over 2 batches under 0, 50, and 100% strain. Results from literature were extracted from corresponding papers. **e** rS and corresponding crack onset strain of PSCs. The rS is defined to quantify the stretchability of PSCs. Additional reported PSCs were analyzed, including PDPPTVT

($M_n$: 20.4 kg mol$^{-1}$)[16], PDPPTVT-PDCA ($M_n$: 16.2 kg mol$^{-1}$)[16], PDPPTVT-alkyl ($M_n$: 24.0 kg mol$^{-1}$)[16], P2TDPP2TFT4 ($M_n$: 29.5 kg mol$^{-1}$)[15], IDTBT ($M_n$: 109 kg mol$^{-1}$)[17], and PII2T-C8 ($M_n$: 434 kg mol$^{-1}$)[18]. **f, g** Correlations between charge carrier mobility and rS values. Normalized FET mobility under **f** 50 and **g** 100% uniaxial strain (i.e., mobility under 50 or 100% strain divided by the mobility at 0% strain) of the studied and reported PSCs are depicted. Normalized mobility for P1–P4 were averaged from at least 5 devices over 2 batches. Results from literature were extracted from corresponding papers based on reported data. A clear trend shows that the electrical performance can be sustained under strain if the PSC possesses a higher rS value. **h** Comparison of rS values between neat PSC (i.e., P4), terpolymer (DPP-8TVT in ref. [19]), and polymer/additive blends (DPP/DOP-r in ref. [25]). rS values for P4 were averaged from at least 2 DR and 2 rDoC samples over 2 batches under 0, 50, and 100% strain. Results of DPP-8TVT and DPP/DOP-r were extracted from corresponding papers.

**Table 1 | Summarized electrical, morphological, and mechanical properties of P1–P4 thin films**

| Polymer | Mn[a] (kg mol⁻¹) | PDI | Mobility[b] (cm²V⁻¹s⁻¹) | On/off ratio[b] | Threshold voltage[b] (V) | rDoC[c] | Crack onset strain (%) | Elastic modulus (MPa) |
|---|---|---|---|---|---|---|---|---|
| P1 | 22 | 1.71 | 1.12 ± 0.034 | $5 \times 10^3$ | −4 | 1 | 3 ± 1 | 347 ± 21 |
| P2 | 50 | 1.96 | 1.38 ± 0.032 | $2 \times 10^4$ | 2 | 0.62 | 25 ± 5 | 109 ± 10 |
| P3 | 73 | 2.21 | 0.932 ± 0.056 | $1 \times 10^4$ | 2 | 0.30 | 85 ± 8 | 47 ± 5 |
| P4 | 97 | 2.11 | 0.550 ± 0.021 | $2 \times 10^5$ | 2 | 0.18 | >130 ± 5 | 23 ± 4 |

[a]Number-averaged molecular weight measured by gel permeation chromatography (GPC).
[b]The electrical performances are averaged from at least 10 devices of 2 different batches.
[c]The rDoC values of the studied polymers are normalized to the crystallinity of P1.

strain, while taking into account their molecular and morphological characteristics.

Upon stretching, mechanical energy can be dissipated by one or more of the following mechanisms: (i) elastic or plastic deformation of the amorphous domains;[47] (ii) molecular alignment or re-orientation of crystallites;[48–50] (iii) amorphization of crystalline domains[51] and (iv) ultimately bond breakage and crack formation. Having multiple possible modes of energy dissipation can potentially produce more stable electrical performance under repeated strain cycles. The change in rDoC and DR are two parameters that can be used to characterize changes in the film due to energy dissipation mechanisms (i)-(iii) during strain. Specifically, the change of rDoC represents whether there is amorphization of ordered domains under strain, while the DR reflects the ability of polymer chains (in both crystalline and amorphous domains) to be deformed and aligned along the strain direction (Fig. 1b)[52]. The details of DR and rDoC measurements will be described later. Interestingly, we found that the ratio between the change of polymer chain alignment measured by DR over the change of rDoC under strain correlated well with both the crack onset strain and reversibility of mechanical and electrical properties of the film (Fig. 1d). Therefore, we define this ratio as the relative stretchability (rS) to capture the contributions from both entropic energy dissipation and enthalpic energy dissipation on the ability of the polymer to tolerate strain. The rS is calculated as the slope of a linear fit of ΔDR as a function of ΔrDoC under 0, 50, and 100% strain. In the DPP-based polymer family, PDPPTVT and PDPPTVT-alkyl[16] thin film are brittle with high modulus, showing a small rS of 0.38 and 0.45, respectively. P2TDPP2TFT4[15] and PDPPTVT-PDCA[16] are both ductile and can be stretched up to uniaxial strain of 50% and 100% without cracks, giving rS of 2.04 and 2.65, respectively. In addition, other stretchable PSCs, such as indacenodithiophene–benzothiadiazole (IDTBT)[17] and isoindigo-bithiophene (PII2T-C8)[18] have rS values of 2.17 and 1.83, respectively. Their rS values are consistent with the strain-dependent thin film properties across polymer structures. For the PSCs in this study, the rS values were calculated to be 1.19, 2.78, 4.16, and 11.5 for P1, P2, P3, and P4, respectively. In general, we found polymers with a higher rS tend to have improved electronic stability and mechanical durability under strain since more mechanical energy can be dissipated under strain through non-deleterious processes, such as chain alignment or crystal alignment (Fig. 1e). Note that values of crack onset strain may be affected by the sample preparation conditions, measuring conditions, and the calibration methods used. Moreover, rS values, in the range from 0.3-3, were calculated for other reported stretchable PSCs based on published data. These rS values correlated well with the trend of electrical properties under strain as reported in the literature (Fig. 1f, g)[15–18]. Therefore, rS provides a reasonable comparison of various designs of stretchable PSCs based on their morphological response to strain. Future work will be carried out to collect additional datasets to identify potential limitations and refinement of rS as a parameter for comparing the potential of polymeric materials for flexible and stretchable electronics. The polymer P4 utilized in this study, exhibited an rS as high as 11.5, which is approximately 4 times higher than previously reported stretchable PSCs, implying a much higher mechanical resilience. In addition, the rS values of not only neat PSCs, but other high-performance stretchable semiconducting polymer systems, such as terpolymers[19] and polymer/additive blends[25], have been analyzed (Fig. 1h). The terpolymer, DPP-8TVT, has an rS values of 7.1, which is higher than most of the reported neat PSCs because of multi-scale molecular ordering by randomizing the conjugated polymer backbone. Additionally, the normalized mobility under strain of DPP-8TVT FETs with a rS value between 5 and 10 followed the general trend of other neat PSCs with similar rS values (Supplementary Fig. S10), suggesting the performance of P4 is not an unexpected outlier. Interestingly, the stretchable polymer with a plasticizing additive (DPP/DOP-r) exhibited a negative rS value because of its strain-induced crystallization (increase of rDoC) under strain. With an absolute rS value larger than 3, these PSC thin films are all ductile and can be deformed uniaxially up to 100% strain without cracking, which is consistent with our observations of the neat PSCs described above. Surprisingly, P4 still showed the highest rS value, a result of the presence of multi-modal mechanical energy dissipation mechanisms under strain. Indeed, P4 thin films can maintain their electrical performance while being expanded 400% in area under a biaxial strain (Fig. 1b), which has never been achieved for any other neat PSC.

## Characterizations of molecular ordering under an external strain

As noted above, molecular conformation and crystallinity play important roles in stretchability of PSCs. A 4-fold increase of rS value was seen for the same polymer chemical structure by simply changing the molecular weight. Grazing-incidence X-ray diffraction (GIXD) was utilized to characterize the evolution of overall molecular ordering (including rDoC, coherence length, and packing orientation) of the PSC films under strain. rDoC values under different strain levels were determined by integrating the intensity of the lamella diffraction peak, taken from 2D diffraction patterns collected with the PSC film rotated with respect to in-plane azimuthal angle φ from 0° to 90°, as illustrated in Fig. 2a[53]. The effect of applied strain on thin film crystallinity is depicted in Fig. 2b. The rDoC value for each polymer was normalized based on its DoC at 0% strain. In general, rDoC decreased as the polymer films were deformed, indicating amorphization of crystallites by tensile strain[51]. As mentioned earlier, strain-induced crystallization is a common phenomenon for many polymers[25,38,39]. Here, our polymer may also have strain-induced crystallization, but the observed overall change of rDoC is a combination of both strain-induced crystallization and strain-induced amorphization with the later process more dominant. The cause for the more dominant strain-induced amorphization in these conjugated polymers may be attributed to the high rigidity of their structures, which may assert a higher stress on the crystalline regions under strain compared to more flexible non-conjugated polymers. The rDoC dropped by ~40% under 50 (P1 film) and 100% (P2 film) strain, respectively, and crystallinity did not recover once strain was released. In addition, a drop in the lamella coherence length was observed in thin films of low molecular weight polymers (i.e., P2) under strain (Fig. 2c). The decrease of the coherence length

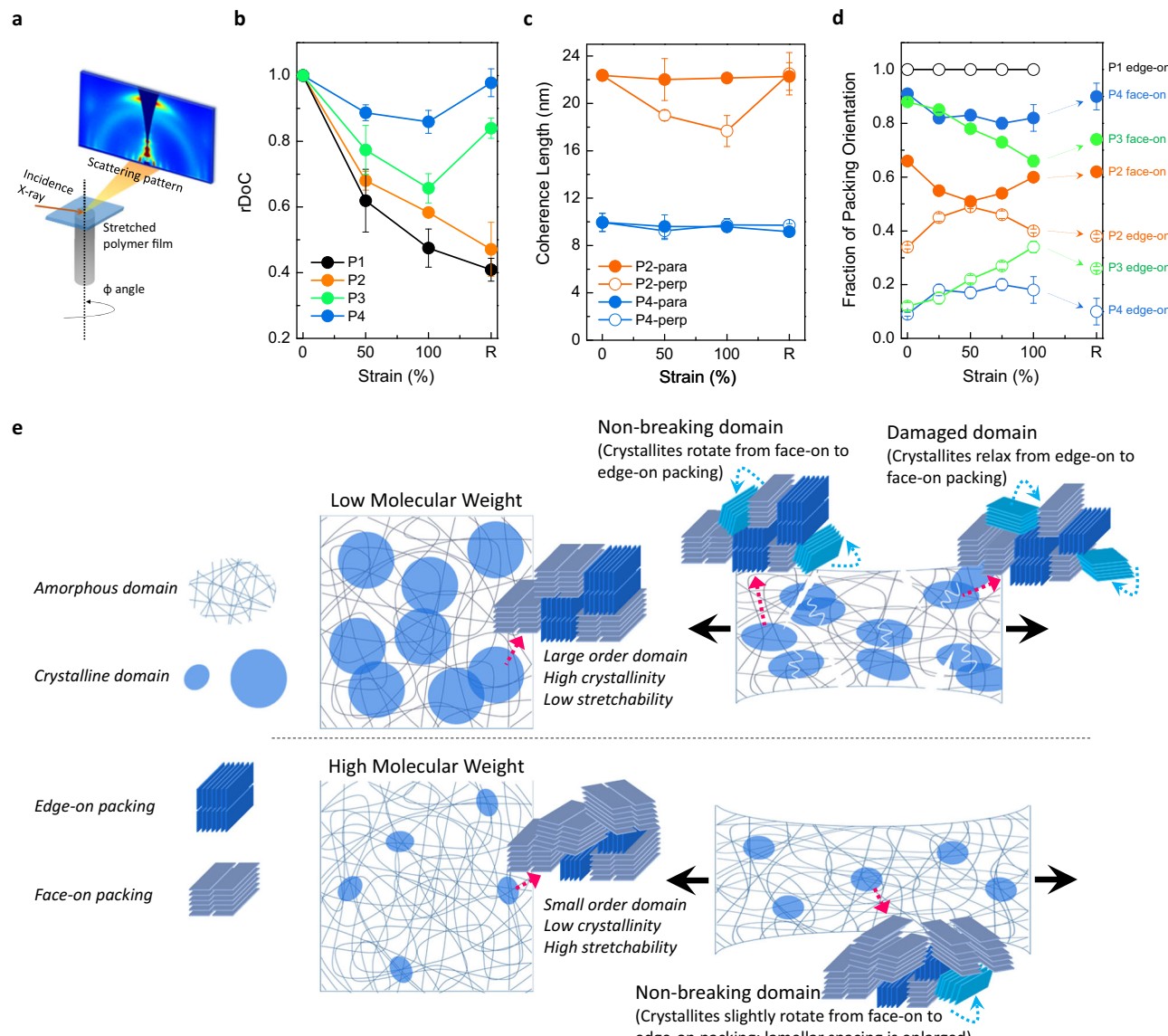

**Fig. 2 | Molecular ordering of PSC films under strain. a** Illustration of GIXD measurement of a stretched PSC film. The sample stage can be rotated to define the direction of the incident X-ray beam with respect to the stretching direction (angle $\varphi$) from $\varphi = 0$ (i.e., X-ray beam is perpendicular to the strain direction) to 90° (i.e., X-ray beam is parallel to the strain direction). **b** rDoC of P1–P4 films as a function of strain. We note that for these incident directions, the scattering vector is not quite parallel (perpendicular) to the strain directions; however, the impact of this limitation is small as the Bragg angles are small for the lamella peak. The crystallinity value is integrated from the diffraction signal in GIXD patterns and is normalized to the initial (0% strain) state for each polymer. **c** The strain-dependent coherence length (i.e., mean crystal size) and **d** packing orientation of P1–P4 films. Edge-on and face-on fractions are defined as the integrated intensity of the (010) reflection from $\chi = 0$–45° and $\chi = 45$–90°, respectively. Significant changes of molecular organization are shown under strain for polymers of various molecular weights. The results shown above were averaged from at least 2 samples over 2 batches. **e** Schematic illustration of molecular dynamics of high and low molecular weight films under strain. The impact of molecular ordering on the resultant mechanical behavior is depicted.

(approximately 20% at 100% strain) was particularly apparent in the direction perpendicular to the strain, suggesting that ordered domains can also break due to compression orthogonally to the stretching direction. In comparison, films based on high molecular weight polymer (P4) showed nearly constant mean coherence length and a much smaller decrease (~10% under 100% strain) in rDoC under strain. Moreover, the rDoC recovered to ~95% of the initial value after strain was removed, which suggests that the crystalline domains can almost recovery reversibly. This reversibility is unexpected for PSCs due to their high rigidity. This behavior was observed for thermoplastic elastomers, such as poly(styrene-ethylene-butylene-styrene) (SEBS), where it has been found that the glassy polystyrene domains exhibit an affine deformation under strain and recover when strain is released[41,54].

In addition, the relative degree of aggregation (rDoA), extracted from the ratio of (0–0) and (0–1) peaks in UV-vis spectra, as a function of applied strain were evaluated (Supplementary Fig. S11) to take into account the behaviors of both longer and shorter range ordered polymer aggregates under deformation since rDoC only depicts long-range order. In contrast to the significant rDoC drop, only small rDoA changes were observed for low molecular weight polymers (i.e., P1 and P2) under strain. This may be due to the polymer films cracking at low strains and the subsequent energy dissipation occurring mainly through crack propagation. Conversely, the high molecular weight P3 and P4 showed less decay in rDoC but more decrease in rDoA because of the continuous film deformation. Additionally, higher values of DR under strain were observed for higher molecular weight P3 and P4.

This is expected as there is a decreased fraction of crystalline domains with molecular weight and higher free volumes of polymer chains in the amorphous regions, both of which provide sufficient flexibility for the polymer chains to undergo conformational extension and alignment under an external strain (Supplementary Fig. S12). This appears reversible for P2–P4, suggesting there is sufficient entropic gain even for these rigid polymers to return to disordered states. Additionally, little degradation in electrical performance for P3 and P4 films under uniaxial strain was observed (Supplementary Figs. S13 and 14). Finally, PSCs with higher $rS$ values exhibited more stable electrical performance under strain (Fig. 1f, g).

Although the strain-dependent rDoC values showed a good correlation between mechanical and electrical properties of semiconducting polymer films, it does not fully take into account all energy dissipation mechanisms. For example, the change of crystalline domain orientation under strain is another mechanism. This aspect was investigated for P1–P4 using 2D GIXD patterns (Fig. 2d, Supplementary Figs. S15–18). The crystalline domains in P1 film may be changed as cracks readily formed. At the same time, there may be a new polymorph formed by applied strain (Supplementary Fig. S19). The higher molecular weight P2–P4 exhibited strain-dependent crystalline domain reorientation. Specifically, P2 initially had 34% and 66% edge-on and face-on packing (Fig. 2d), respectively. When a tensile strain was applied, the fraction of edge-on crystallites increased to nearly 50%, at 50% strain, indicating that the crystallites can be rotated or tilted, which is another possible mechanism for strain energy dissipation[48,55]. Such a reorientation is expected to give some degree of alignment, which is accounted for by DR values. However, the edge-on packing fraction of P2 decreased back to 40% under 100% strain, which could be due to micro-scale crack formation and released stress, as observed by microscopic images (Supplementary Figs. S6 and 7). Similarly, a monotonic increase in edge-on packing with strain was observed for polymers with higher molecular weight, especially with P3 film. Without microcracking (Supplementary Fig. S6b), the face-on and edge-on ratio continued to change and contribute to energy dissipation up to 100% strain. Note that P4 films exhibited lower levels of changes of the packing orientation, probably due to its low DoC (Supplementary Fig. S5).

Based on the above observations, the molecular picture of the PSC films under strain can be summarized as shown in Fig. 2e. The lower molecular weight polymer films initially possessed larger mean crystallite sizes and greater rDoCs. The crystallites rotate under a low external strain in addition to chain extension and alignment in the amorphous regions. Some aligned chains may undergo strain-induced crystallization, even though this is minimal in P1–P4 systems. At a high strain level, crystallites may be amorphized and eventually cracks form. In contrast, thin films of higher molecular weight polymers tend to have smaller crystalline domain sizes and lower rDoC. A larger fraction of molecules is in the disordered amorphous phase and have more freedom to reorient, change conformation and align under applied strains, leaving the small fraction of small crystallites mostly intact. Moreover, the stress distribution in stretched thin films with crystalline domains can be visualized by a finite element simulation (Supplementary Fig. S20). It can be seen that stress tends to be more concentrated in the larger crystalline domains, while large strain concentrations are observed in the lower molecular weight polymer films. Such a morphology is more likely to experience damage (i.e., micro-cracks), as observed in Supplementary Fig. S6. On the other hand, high molecular weight films possess smaller crystallites with low density. Thus, stress is distributed more uniformly and is less localized on the crystalline domains. The above molecular picture shows the way to useful design rules for intrinsically stretchable PSCs. For example, a terpolymer of multiple conjugated building blocks, which conforms to these design criteria, has been reported to exhibit exceptional electronic performance under tensile strain[19]. Again, both

molecular alignment and crystallite re-orientation are characterized by DR, while the crystallite fraction and amorphization are characterized by rDoC. Therefore, $rS$ (i.e., $\Delta DR/\Delta rDoC$; entropic energy dissipation/enthalpic energy dissipation) takes into account the above-described major factors contributing to thin film stretchability (recall Fig. 1d; P4 thin film with $rS$ value of 11.5 is the one with the highest crack-onset strain).

## Stretching thin films under multi-directional strain

P4 is a promising candidate for stretchable electronics, which require the active semiconducting layer be subjected to non-uniform stress (stretching in any combinations of directions). So far, most reported works have only cover investigations of PSC films under a uniaxial strain. Here, simultaneous biaxial stretching of the polymer films is studied as it is closer to realistic conditions (details of the biaxial stretching process are illustrated in Supplementary Fig. S21). Figure 3a shows the AFM surface topography of P2 and P4 films under biaxial 100% strain. Remarkably, even under 100% strain, which corresponds to a 400% expansion by area, the P4 film remained smooth and no cracks could be observed. In comparison, P2 films were completely damaged by the biaxial stretching and micro-scale cracks were readily observable (Supplementary Fig. S22). In addition, the rDoC and coherence lengths of such films under biaxial strain were extracted from 2D GIXD patterns (Supplementary Fig. S23). Similar to the results of uniaxially stretched films, lower molecular weight P2 films showed substantial decrease in rDoC and coherence length under biaxial strain, indicating the crystalline domains in the film were mostly amorphized or cracked. The molecular ordering in high molecular weight P4 thin films, however, appears to be maintained even under a high level of strain. The rDoC decreased by less than 25% and the mean crystal size stayed almost the same as its initial (i.e., non-stretched) state under 100% biaxial strain.

Since P4 films have the unusual capability to resist damage from applied strain during biaxial stretching, a stable charge transport performance was obtained (Fig. 3b). P4 maintained nearly constant field-effect mobility of 0.15 cm$^2$V$^{-1}$s$^{-1}$ with negligible hysteresis under biaxial strain of 0–100% (Supplementary Fig. S24). Effects of biaxial strain on organic electronic materials have rarely been investigated[28,56]. This is the first report of a PSC achieving stable electrical behavior with a mobility greater than 0.1 cm$^2$V$^{-1}$s$^{-1}$ under biaxial deformation. Previously, only (i) P3HT was stretched biaxially using a two-step method (stretched one direction to 100% and then stretched the orthogonal direction to 100%) with a resulting mobility lower than 0.01 cm$^2$V$^{-1}$s$^{-1}$ under biaxial strain[56], and (ii) a P3HT/polydimethylsiloxane (PDMS) blend exhibited a dramatic decay in mobility (over 85%) under 50% biaxial strain[28] (Supplementary Fig. S25), were reported. As expected, compared to P4, the mobility of P2 and P3-based films exhibited significant decay upon biaxial stretching. The corresponding stable (P4-based device) or unstable (P2-based device) current output as a function of strain is depicted in the FET transfer characteristics (Fig. 3b and Supplementary Fig. S26).

Furthermore, high operational cycling stability of the stretchable P4 film was observed. Almost unchanged electrical performance was observed after 500 stretch and release cycles of 100% biaxial strain, as shown in Supplementary Fig. S27. In addition, a 3 × 3 array of electrodes with different in-plane orientations were used to study the anisotropy of charge transport in biaxially stretched films (Supplementary Fig. S28). The on-state currents of FETs along various charge transport directions with respect to the stretching direction were observed to be approximately the same, indicating isotropic charge transport in biaxially deformed films. Additionally, the effect of anisotropic biaxial stretching (i.e., 25 × 50%, 50 × 75%, and 50 × 100%) was investigated. Unlike the symmetric biaxially stretched films, which did not experience force-induced polymer chain alignment due to the isotropic force applied, films under anisotropic biaxial strain exhibited DR values

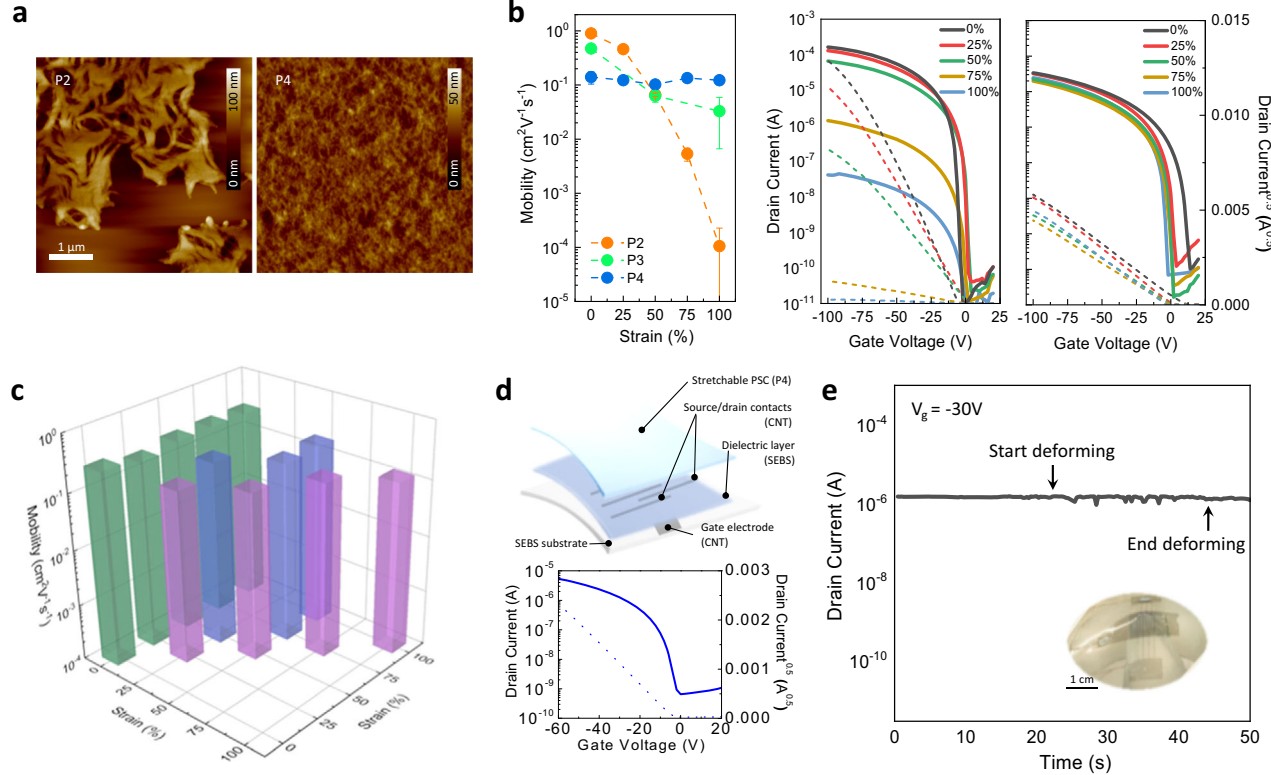

**Fig. 3 | Simultaneous biaxial stretching of PSC films. a** AFM topographies of P2 and P4 film under biaxial 100% strain. With suitable molecular ordering and good intrinsic stretchability, the P4 film shows a smooth and uncracked surface even under 400% area expansion. **b** Averaged FET charge carrier mobilities (for at least 5 devices from 2 batches) and representative transfer curves of PSC films under biaxial strains. The source-to-drain voltage was set as -100V. A dramatic decay of electrical properties is observed for the low molecular weight polymer (i.e., P2), while P4-based FETs exhibit stable and superior electrical performances under strain. **c** Comparison of charge carrier mobility (averaged from 5–10 devices over 2 batches) of P4 films under uniaxial or biaxial strain. Conditions of even (e.g.,

50 × 50%, 100 ×100%, etc.) or uneven (i.e., 25 × 50%, 50 × 75%, and 50 ×100%) biaxial strains are included. Comparable mobility of P4-based FET is observed in the range of 0.1–0.25 cm²V⁻¹s⁻¹, suggesting a stable electrical performance will be maintained, even if the film is randomly deformed. **d** Device configuration and representative electrical characteristics of a fully stretchable FET device based on P4 thin films. The charge carrier mobility of the fully stretchable device is comparable to the devices fabricated on rigid Si substrates. The source-to-drain voltage was set as −60V. **e** Operational stability of a fully stretchable device under irregular strains by poking with a 15 mL plastic centrifuge tube in a circular motion. Stable current output is observed under a reading gate voltage of −30V.

larger than 1 along the direction with a larger strain, indicating alignment along the larger strain direction (Supplementary Fig. S29). However, the degree of polymer alignment in this case was much smaller than that from uniaxial stretching because part of the tensile strain is canceled from the compression generated in the perpendicular direction due to the Poisson effect. For the P4 film stretched to 50 × 100% strain for 50 stretch/release cycles, the DR stayed in a range of 0.95–1.2 (Supplementary Fig. S30). As a result, we observed that the charge carrier mobility changed when an anisotropic biaxial strain was applied (Supplementary Table S1). Along the direction with a higher net strain, both the mobility and DR were slightly higher.

Comparing P4 FET devices exposed to a uniaxial, isotropic biaxial, or anisotropic biaxial strain, consistent mobilities between 0.1–0.25 cm²V⁻¹s⁻¹ were achieved, suggesting that no matter which direction the force was applied to the PSC film, excellent maintenance of charge transport was observed (Fig. 3c).

The above observation is new and surprising for rigid conjugated semicrystalline polymer semiconductors. Such mechanical behavior is only observed previously for conventional rubber materials, in which the material can be anisotropically strained and return to its original shape[41,57]. Remarkably, P4 can achieve reversible behavior (correlated with its high *rS* value). As discussed above, reversible chain alignment, chain and crystallite re-orientation, and crystallite amorphization and reformation are important molecular mechanisms for energy dissipation. In this case, all the above mechanisms can be contributed by

tuning the molecular weight of the conjugated polymers despite their rigid polymer backbones.

## Discussion

Our work above suggests *rS* is a relevant parameter for comparing promise of PSCs for stretchable FETs. Both amorphous and crystalline domains may dissipate the accumulated stress in stretched thin films. A higher *rS* value is obtained when there is a greater energy dissipation during film deformation, resulting in reversible molecular ordering, as observed in various PSC systems (Supplementary Fig. S31). Similar to thermoplastic elastomers, such as SEBS, the crystallinity of the smaller, more rigid domains are more likely to be maintained during deformation as less stress is concentrated on smaller hard domains distributed in larger soft amorphous domains, as compared to larger hard domains distributed within smaller soft domains[41,54]. In general, the *rS* takes into account energy dissipated entropically through polymer conformation change and enthalpically through amorphization of crystallites, respectively, which can quantitatively describe the strain-dependent behaviors of semicrystalline stretchable PSCs. Finally, a fully stretchable FET device using high *rS* P4 as the active material was fabricated[30] and a charge carrier mobility of approximately 0.2 cm²V⁻¹s⁻¹ was measured (Fig. 3d). Operational stability of such a fully stretchable device under multi-directional deformation is depicted in Fig. 3e. Negligible degradation was observed in source-drain current before and

after stretching, suggesting the electrical performance can be maintained under irregular strain.

In summary, the ability for reversible molecular ordering under strain is found to be a key factor in realizing robust mechanical and electrical performance of intrinsically stretchable PSCs. We found that relative stretchability (rS) correlated well with a polymer's ability to align and maintain crystalline domains as an indication of the effectiveness of stress energy dissipation of the conjugated PSC films under strain. Indeed, PSCs possessing high rS showed reversible molecular ordering, which is similar to broadly studied semicrystalline non-conjugated rubbers whose glassy domains can be deformed reversibly with small domain sizes. Our findings provide a way to compare stretchable polymer semiconductor materials.

## Methods
### Polymer characterization
The number-averaged molecular weight ($M_n$) and polydispersity index (PDI) of these polymers were measured by high temperature size exclusion chromatography (SEC) (Polymer Labs (Now Owned by Agilent) GPC 220 system with a refractive index detector) at 200 °C using polystyrene as molecular weight standards and 1,2,4-trichlorobenzene as eluent. Dynamic mechanical analysis (DMA) was performed on a TA Instrument Q800 with a static force of 0.01 N, an oscillation strain of 0.1% at 1.0 Hz and a ramp rate of 2 °C min$^{-1}$. A PSC solution (10 mg mL$^{-1}$ in chlorobenzene) was drop-casted on top of the polyimide film and the films were then annealed at 100 °C for 3 h to relax the polymer chains. Afterwards, the polyimide film coated with PSC was removed from the supporting OTS-modified Si substrate and installed between the DMA clamps for measurement. The polyimide films and sample preparation were followed by our published procedure[58]. The mechanical strain-stress tests were performed by using an Instron 5565 instrument with a stretching rate of 10% s$^{-1}$. A PSC solution (10 mg mL$^{-1}$ in chlorobenzene) was drop-casted on top of a Teflon plate in the ambient environment and annealed at 100 °C for 1 h. The thickness of the drop-casted films is in a range of 2–3 μm. Finally, the free-standing PSC bulk films were installed between the Instron clamps for measurement.

### PSC thin film characterization
The elastic modulus of PSCs was determined using the buckling method[59]. The polymer films of approximately 50 nm were spin-coated on an OTS-modified Si substrate and annealed at 100 °C for 20 min, then the PSC thin films were transferred onto a pre-strained (i.e., 4%) PDMS substrates. Once the strain was released, wrinkles were clearly formed. The elastic moduli of the films were calculated from the wrinkle wavelength and film thickness. The surface morphology of PSC thin films was recorded by a Nanoscope 3D controller atomic force micrograph (AFM, Digital Instruments) operated under tapping mode at room temperature. Optical images of the PSC films were taken by an optical microscope (Leica DM4000M). Grazing-incidence X-ray diffraction (GIXD) experiments were performed at beamline 7-2 and 11-3 in Stanford Synchrotron Radiation Light source (SSRL) with a photon energy of 14 and 12.73 keV, respectively. Note that the incident angle was fixed at 0.12° to improve the diffraction intensity and reduce the substrate scattering. All GIXD images were collected in reflection mode with a 2D area detector, and the samples were placed under a Helium atmosphere. Data analysis was performed in Wixdiff, written by S. Mannsfeld. The relative degree of crystallinity (rDoC) values under different strain levels were determined by integrating the intensity of the lamella diffraction peak, taken from 2D diffraction patterns collected with the PSC film rotated with respect to angle φ from 0° to 90°. The integrated intensity was normalized by the scattering intensity to scattering volume and exposure time. Additionally, the scattering intensity was corrected for scattering geometry, as detailed in previous work by Toney and co-workers[60] and a sin(omega) correction was applied for the data analyses. Agilent Cary 6000i UV/Vis/NIR spectroscopy was used to measure the UV-vis absorption spectra. A polarizer crystal was equipped to measure the absorption intensity with the polarization parallel ($A_{//}$) and perpendicular ($A_{\perp}$) to the stretching directions. The dichroic ratio (DR) was determined by $A_{//}/A_{\perp}$ for analyzing polymer chain alignment under strain. The rS value is defined as the ratio of DR/rDoC at various strains, the DR values were averaged from at least 3 samples from 3 different batches and the rDoC values were averaged from at least 3 samples from 3 different batches for each polymer under a specific strain condition (i.e., 0, 50, or 100% strain). All the FET devices were probed using a Keithley 4200 semiconductor parameter analyzer (Keithley Instruments Inc.) in ambient at room temperature

## Data availability
The authors declare that all data supporting the findings of this study are available within the article and its Supplementary Information files or from the corresponding author upon request.

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

## Acknowledgements

This work is supported by the Air Force Office of Scientific Research under award numbers FA9550-18-1-0143 (17RT0917) and FA9550-21-1-0413 (21RT0491). J.X. acknowledges the Center for Nanoscale Materials, supported by the U.S. Department of Energy, Office of Science, under Contract No. DE-AC02-06CH11357. N.M. acknowledges funding support from an overseas fellowship from the Japan Society for the Promotion of Science (JSPS). C.L. acknowledges support from the National Science Foundation through grant CMMI-1911836. This work was partially performed at the Stanford Nano Shared Facilities (SNSF), supported by the National Science Foundation under award ECCS-1542152. GIXD experiments were carried out at the Stanford Synchrotron Radiation Laboratory (SSRL), a national user facility operated by Stanford University on behalf of the U.S. Department of Energy, Office of Basic Energy Sciences. The authors acknowledge Bart Johnson at SSRL for supporting GIXD experiment and Brandon Clark for helping thin film characterizations.

## Author contributions

H.-C.W and S.N. contributed equally to this work. H.-C.W., M.H., and Z.B. conceived and designed the experiments; H.-C.W., S.N., C.Z., N.M., J.X., and Y.-Q.Z. fabricated and characterized thin films and transistor devices; H.-C.W., S.N., and H.Y. did the GIXD characterizations; Y.L., W.N., J.R.M. and M.H. designed and synthesized the conjugated polymers; P.K.A., R.R., and C.L. carried out the simulation of stretched films; H.Y. and M.F.T. advised on the discussion of morphological results; H.-C.W. organized the data and wrote the first draft of the manuscript. All authors reviewed and commented on the manuscript. Z.B. directed the project.

## Competing interests

The authors declare no competing interests.
