## [Peer Review File · Nature Communications]

REVIEWER COMMENTS

Reviewer #1 (Remarks to the Author):

The manuscript has been heavily revised in response to my comments and those of the other reviewers. It is improved in its presentation and contextualization, but I still do not agree entirely with the proposal for a new dimensionless number. I would encourage the editors to continue to consider the manuscript for publication in this fine journal provided the following points are considered.

Re Q1: The presentation of the materials suite is now excellent, thank you.

Re Q2: Slightly more context has been added for why a new figure of merit might be required, most importantly the assertion that the phenomenology being characterized is specifically the behavior of thin films on elastomeric supports. However, I still don't agree that the proposed dimensionless number is scientifically suitable. I'll address rS more below.

Re: Q3: I'll address the scientific suitability of rS below.

Re Q4: the authors are correct that many traditional figures of merit for elastomers are assessed in a more bulk-like state and might be difficult to assess for thin films. Defects will spoil many of the measurements. The changes to the title and the text to emphasize the new material rather than the figure of merit improve the manuscript. I'll address rS more below.

Re Q5: The authors incompletely address my concerns about strain-induced crystallization being the most typical response to strain in semicrystalline polymers. They present four references to support the notion of strain-induced amorphization; again, I would contrast this with thousands of references in the open literature on strain-induced crystallization of semicrystalline polymers. Of the four references, all are semiconducting polymers, and three out of the four references are from the Bao lab. The fourth reference *Macromolecules* 49, 8540-8548 (2016), the only non-Bao-lab reference provided, does not appear to provide evidence for strain-induced amorphization and as far as I can tell that paper does not even calculate $rDOC$, so it should be removed from the block of references that support the prevalence of strain-induced amorphization.

We are left with citations that support 1) strain-induced amorphization appears unique (?) to polymer semiconductors, and 2) strain-induced amorphization has been reported exclusively by the Bao lab, at least in the references provided. To be clear, the Bao lab does great work and I completely trust that this phenomenon is happening, but I am lingering on this topic to make a point about how the phenomenon is contextualized. In lines 81-82, where strain-induced amorphization is introduced, it would be best to indicate that the most common result of strain in semicrystalline polymers is additional crystallization,

and that, curiously, both strain-induced crystallization and strain-induced amorphization have both been observed in polymeric semiconductors.

Re Q6: The authors incompletely address my concern about T_g . I agree that crystallinity and molecular mass affect elastomeric properties but to observe rubbery behavior at room temp requires T_g below room temp. Reviewer 2 Q7 echoes my concern. Perhaps a statement within the main text speculating that the thin-film T_g differs from the bulk T_g would reconcile the observations?

Re rS: Several of my concerns relate to the necessity and scientific suitability of the proposed new dimensionless number. The revised manuscript improves the rationale, but it still doesn't completely address my core concern in Q3, which relates to why these two measurements should be meaningfully related to one another and why their ratio should provide meaningful information across a wide range of materials.

There are no "rules" about the formulation of dimensionless numbers across the scientific enterprise, but there are some commonalities among dimensionless numbers that have been broadly adopted. Many dimensionless numbers are ratios of quantities that have the same meaningful units – usually forces – that stem from different physical phenomena that affect the same body or medium. Consider the Reynolds number, the ratio of inertial force to viscous force. Most fluids dynamics numbers are similar ratios of forces. There is a different category of dimensionless numbers that are coefficients such as coefficients of friction. In most cases, dimensionless numbers also fall out of classic equations, just as the Reynolds number naturally falls out of relationships between fluid velocity and pipe diameter.

The proposed quantity, rS, is not a ratio of quantities with meaningful units. Although it is intended to evaluate different mechanisms of energy dissipation under strain, it is not proposed as a ratio of energies. It is instead proposed as a ratio of two other measurements, dichroic ratio and rDOC loss, the numerical values of which have very different meanings. rS also does not naturally derive from any equations.

If the authors are intent on pursuing rS, I would suggest elaborating on their concept by formulating it more rigorously as a ratio of energies: specifically, the ratio of energy dissipated by non-deleterious processes (DR, reorientation) vs. energy dissipated by deleterious processes (amorphization). Perhaps they could even preserve their core ratio, but Δ -DR would need to be adjusted by viscoelastic properties (the energy dissipated by the same reorientation will be different if viscosity is different), and Δ -rDOC would need to be adjusted with latent heat of fusion and somehow asserted on a more absolute basis. For the paper under review, perhaps it would be enough to assert that viscoelastic properties and heat of fusion are assumed to be similar across the series for a preliminary assessment of such a quantity. This approach would preserve the nice correlation the authors report, and also make the quantity more future-proof if systems are found that have very different viscoelastic properties or heats of fusion that would break the trends. Perhaps it could even extend in the future to add additional deleterious energy dissipation mechanisms such as crack formation and propagation.

Reviewer #3 (Remarks to the Author):

The authors have revised this paper taking into account all my suggestions. Furthermore, the paper was revised to enhance clarity of the presentation of significance to the field of thin-films for TFT/optoelectronics. I believe this work can be published without additional modifications.

Reviewer #4 (Remarks to the Author):

The manuscript introduces a DPP-based highly stretchable polymer semiconductor (P4) that can be biaxially stretched up to 100% strain with stable electrical performance by dissipating the energy through multiple mechanisms. The authors develop a new metric, relative stretchability (r_s), to quantitatively evaluate the potential for polymer semiconductors (PSC) in stretchable electronics, especially for thin film transistors. To prove that PSC with high r_s has better and reversible performance, the authors reported a new polymer with a series molecular weights (P1-P4). I believe the P4 polymer has quite impressive performance and shows great potential for the stretchable transistors. Also, it is important to have a new metric to evaluate the PSC that can incorporate existing parameters, like elastic modulus, the crack on set strain (COS), crystallinity and others. However, there is not enough evidence to prove that r_s can more accurately predict the performance of PSC. The idea is good, but I have questions and concerns about validation of this metric, specifically its generality. I would recommend this for publication if the concerns and the questions are properly addressed.

Q1. In page 5 line 112, reference 37, 40, and 41 are irrelevant to the T_g , if the references are mislabeled, please correct them.

Q2. In the abstract and in page 8, the authors claim that “PSC with low r_s value showed low crack onset strain” and “a larger r_s value indicates more mechanical energy can be dissipated under strain through chain alignment or crystal alignment and corresponding a high crack onset strain (COS) and more reversible electrical performance of the polymer” from Figure 1e. The r_s of P1-P4 has a good correlation with the COS, however, it is not the case for other polymers. This r_s and COS correlation seems to work the best for the designed PSC but with different Mw, when the backbone and side chain are different, it loses the generality. For DPP-based polymer, P1 has much higher r_s value but significant lower COS than PDPPTVT, not mention PDPPTVT-alkyl which has more modifications on the backbone. There are some limitations to compare PSC across the polymer libraries with this metric. It is necessary to prove generality for a new proposed metric. There are plenty of commercially available and lab reported materials for thin film transistors, but how this metrics work for these other materials is currently an unknown.

Q3. In Figure 1 f and g, the correlation of normalized mobility and rS seems good at low rS value. As there is no data point in the middle range (5-10) and P4 appears to be an outlier when compared to the other points. It would help to better describe the trend to have a few data points to cover the whole rS range if possible (from 0 to 10). Doing so would show that while P4 appears to be off trend, it is properly represented.

Q4. It is unclear whether the mobility plotted in Figure 1 f and g is parallel or perpendicular to the strain direction or the average of both. From Figure S11, P1-P4 seems to have isotropic charge transportation under strain. Some polymers have anisotropic charge transport under uniaxial strain. How does the mobility vs rS plot of those polymers look in both directions?

Q5. The authors state that rS shows good correlation with the reversibility of electrical performance as well when rS was first introduced in the main text, page 7. The only reversibility I can tell is $rDoC$, the molecule ordering. However, there is no direct evidence which shows that reversibility of the electrical properties and mechanical properties were related to the rS . As all electrical properties (mobility) plots are under strain, when the authors want to claim the reversibility, it will be helpful to have the plot of mobility vs rS at the strain release condition and/or some physical supports.

Q6. A follow-up question for reversibility in mechanical properties. P2, P3 and P4 all have clear yield points from 2% to 5% and P4 has a highest yield stress. With such a clear yield point and a high yield stress, the statement of the mechanical reversibility is surprising. Although the mechanical behavior can change from bulk to thin film, a full cycle of stress-strain curve (before the elongation at break) would back up the statement. At least the curve for P4 polymer should be provided. It will be more necessary to provide the stress-strain curve of the thin film if the authors want to make a point on the mechanical properties of thin film. In addition, no other morphology results except the GIXD results of the strain released films were provided. If the thin film is transferred to a PDMS substrate, a simple microscope image of strain released film should be accessible.

Q7. In page 11, the authors mentioned that the edge on packing with strain is increasing monotonically for P3 and P4, however, in Figure 3d the packing orientations suggests that P3 has monotonic increase in the edge on packing and the edge on packing of P4 appears to plateau from 25% strain with a fraction of 20%. From my point of view, the energy may not be dissipated through the crystallites rotation since 25%. This behavior does not exactly match the statement made in the context. The authors should be cautious when providing an explanation for this observation.

Q8. In Figure 2b, the $rDoC$ of P2 keeps dropping from 60% to 50% when the 100% uniaxial strain is released, but in Figure S21, the $rDoC$ of P2 recovers from 20% to almost 50% when the 100% biaxial strain is released. If the $rDoC$ increased after the strain is released, does that mean the crystalline domain in P2 is recovered under biaxial strain but not under uniaxial strain? In addition, the coherence

length of P2 in perpendicular direction under uniaxial strain is pretty much the same as the coherence length in biaxial strain. The authors used the orthogonal compression to explain the drop in coherence length at 100%, but when the film is uniformly biaxially stretched, what breaks the domain? These behaviors are different from what I would have expected, and I hope the authors can clarify these observation

We appreciate the feedback from the reviewers. Below is a point-to-point response to their comments:
Reviewer #1 (Remarks to the Author):

The manuscript has been heavily revised in response to my comments and those of the other reviewers. It is improved in its presentation and contextualization, but I still do not agree entirely with the proposal for a new dimensionless number. I would encourage the editors to continue to consider the manuscript for publication in this fine journal provided the following points are considered.

1. Re Q1: The presentation of the materials suite is now excellent, thank you.

Response: We thank the reviewer for the kind response.

2. [Comments related to rS]

Re Q2: Slightly more context has been added for why a new figure of merit might be required, most importantly the assertion that the phenomenology being characterized is specifically the behavior of thin films on elastomeric supports. However, I still don't agree that the proposed dimensionless number is scientifically suitable. I'll address rS more below.

Re Q3: I'll address the scientific suitability of rS below.

Re Q4: the authors are correct that many traditional figures of merit for elastomers are assessed in a more bulk-like state and might be difficult to assess for thin films. Defects will spoil many of the measurements. The changes to the title and the text to emphasize the new material rather than the figure of merit improve the manuscript. I'll address rS more below.

Re rS: Several of my concerns relate to the necessity and scientific suitability of the proposed new dimensionless number. The revised manuscript improves the rationale, but it still doesn't completely address my core concern in Q3, which relates to why these two measurements should be meaningfully related to one another and why their ratio should provide meaningful information across a wide range of materials.

There are no "rules" about the formulation of dimensionless numbers across the scientific enterprise, but there are some commonalities among dimensionless numbers that have been broadly adopted. Many dimensionless numbers are ratios of quantities that have the same meaningful units – usually forces – that stem from different physical phenomena that affect the same body or medium. Consider the Reynolds number, the ratio of inertial force to viscous force. Most fluids dynamics numbers are similar ratios of forces. There is a different category of dimensionless numbers that are coefficients such as coefficients of friction. In most cases, dimensionless numbers also fall out of classic equations, just as the Reynolds number naturally falls out of relationships between fluid velocity and pipe diameter.

The proposed quantity, rS, is not a ratio of quantities with meaningful units. Although it is intended to evaluate different mechanisms of energy dissipation under strain, it is not proposed as a ratio of energies. It is instead proposed as a ratio of two other measurements, dichroic ratio and rDOC loss, the numerical values of which have very different meanings. rS also does not naturally derive from any equations.

If the authors are intent on pursuing rS, I would suggest elaborating on their concept by formulating it more rigorously as a ratio of energies: specifically, the ratio of energy dissipated by non-deleterious processes (DR, reorientation) vs. energy dissipated by deleterious processes (amorphization). Perhaps they could even preserve their core ratio, but delta-DR would need to be adjusted by viscoelastic properties (the energy dissipated by the same reorientation will be different if viscosity is different), and delta-rDOC would need to be adjusted with latent heat of fusion and somehow asserted on a more absolute basis. For the paper under review, perhaps it

would be enough to assert that viscoelastic properties and heat of fusion are assumed to be similar across the series for a preliminary assessment of such a quantity. This approach would preserve the nice correlation the authors report, and also make the quantity more future-proof if systems are found that have very different viscoelastic properties or heats of fusion that would break the trends. Perhaps it could even extend in the future to add additional deleterious energy dissipation mechanisms such as crack formation and propagation.

Response: We appreciate the comments and thoughtful suggestions from the reviewer. Although rS is not derived from classical equations, indeed we intended it as a ratio of energies as the reviewer was able to provide a more concise definition: the entropic energy dissipation (polymer conformation change and crystallite re-orientation, which relates to degree of chain alignment as well as DR change, assuming similar viscoelastic properties) and enthalpic energy dissipation (crystallization and amorphization of crystallites, which is correlated to $rDoC$ change assuming similar latent heat of fusion) under strain. Most reported studies used DR, $rDoC$, or crack onset strain to evaluate if a semiconducting polymer is suitable for stretchable transistor applications. However, it is difficult to compare characterization results between different works because each case may involve different processing steps to fabricate stretchable semiconducting polymer thin films (i.e. < 100 nm). For example, high crack onset strain does not necessarily correlate to the ability to maintain electrical performance under strain as electrical performance depends on changes in polymer orientation and microstructures. Some PSCs, such as PDPPTVT-PDCA and IDTBT, exhibited high crack onset strain, but their normalized mobility under strain was relatively lower (Figure R1). When we use DR and $rDoC$ to compare trends, we can eliminate the processing variations since the thin films for DR and $rDoC$ measurements usually are prepared in a similar way, and thus we start to be able to compare the relationship between mechanical and electrical properties of PSCs across different groups, as depicted in Figure 1f.

Figure R1 | Correlations between charge carrier mobility and crack onset strain. Normalized FET mobility under 50 % uniaxial strain (i.e. mobility under 50% strain divided by the mobility at 0% strain) of the studied and reported PSCs are depicted.

Figure 1f | Correlations between charge carrier mobility and rS values. Normalized FET mobility under 50% uniaxial strain (i.e. mobility under 50 or 100% strain divided by the mobility at 0% strain) of the studied and reported PSCs are depicted.

We agree the evaluations of viscoelastic properties and latent heats of fusion may allow the rS to be calculated more accurately. The viscoelastic properties and latent heats of fusion traditionally can be performed on bulk semiconducting polymers or polymer solutions. However, these properties will not be straightforward to measure on a semiconducting polymer thin film of tens of nanometers thick, especially under an applied strain, which is beyond the scope of this manuscript. Furthermore, such properties will be dependent on the detailed morphology, which will be affected by the processing conditions.

Following the helpful suggestions by this reviewer, we have modified the descriptions in the manuscript of rS and present it as a ratio of two energies. rS is a useful tool and parameter for comparing the reliability and reversibility of stretchable semiconductor performance across various types of PSC thin films reported by the research community, and further refinement will be explored in future works.

Changes made:

We have revised the descriptions of the abstract on page 1 to present rS as a ratio of two energies:

“...relative stretchability (rS) defined by the ratio of the entropic energy dissipation (polymer conformation change and crystallite re-orientation, which relate to degree of chain alignment, assuming similar viscoelastic properties) and enthalpic energy dissipation (crystallization and amorphization of crystallites, which is related to the relative degree of crystallinity ($rDoC$) change assuming similar latent heat of fusion) under strain...”

In addition, the descriptions in pages 4 and 8 of the revised manuscript have been modified to present the relationship between rS and energy dissipations:

“...relative stretchability (rS) to compare the relative strain tolerance of various PSC thin films for transistor applications. rS is defined as the change of polymer chain alignment (i.e. dichroic ratio (DR)) over the change of relative degree of crystallinity ($rDoC$) as a function of strain, $rS = \Delta DR / \Delta rDoC$. The degree of chain alignment under strain directly correlates with energy dissipated entropically through polymer conformation change and crystallite re-orientation assuming

similar viscoelastic properties while the change of $rDoC$ ($\Delta rDoC$) represents the amount of energy dissipated enthalpically through amorphization of crystallites assuming similar latent heat of fusion (taking into account there may be crystal formation due to strain-induced crystallization). Therefore, rS presents the ratio of the entropic energy dissipation over the enthalpic energy dissipation....” ; “...we define this ratio as the relative stretchability (rS) to capture both the contributions from both entropic energy dissipation and enthalpic energy dissipation on the ability of the polymer to tolerate strain....”

3. Re Q5: The authors incompletely address my concerns about strain-induced crystallization being the most typical response to strain in semicrystalline polymers. They present four references to support the notion of strain-induced amorphization; again, I would contrast this with thousands of references in the open literature on strain-induced crystallization of semicrystalline polymers. Of the four references, all are semiconducting polymers, and three out of the four references are from the Bao lab. The fourth reference *Macromolecules* 49, 8540-8548 (2016), the only non-Bao-lab reference provided, does not appear to provide evidence for strain-induced amorphization and as far as I can tell that paper does not even calculate $rDOC$, so it should be removed from the block of references that support the prevalence of strain-induced amorphization.

We are left with citations that support 1) strain-induced amorphization appears unique (?) to polymer semiconductors, and 2) strain-induced amorphization has been reported exclusively by the Bao lab, at least in the references provided. To be clear, the Bao lab does great work and I completely trust that this phenomenon is happening, but I am lingering on this topic to make a point about how the phenomenon is contextualized. In lines 81-82, where strain-induced amorphization is introduced, it would be best to indicate that the most common result of strain in semicrystalline polymers is additional crystallization, and that, curiously, both strain-induced crystallization and strain-induced amorphization have both been observed in polymeric semiconductors.

Response: We thank the reviewer for the comments. As noted by the reviewer, strain-induced amorphization seems to be observed in some conjugated semiconducting polymers but not generally observed for non-conjugated semicrystalline polymers. However, we have found studies about strain-induced amorphization from metallic materials (*Phys. Rev. Lett.* 82, 2900 (1999); *Nature* 591, 82-86 (2021); *Appl. Phys. Lett.* 82, 2017 (2003)) and some reports on graphite or ceramics (*Journal of Structural Geology*, 72, 142-161 (2015); *Prog. Mater. Sci.* 112, 100664 (2020)). It appears that the strain-induced amorphization happens when the material generally possesses a high crystallinity. This may explain why strain-induced amorphization is harder to be observed in classical semicrystalline polymers because their crystallinity is lower in nature. It makes sense as low crystallinity polymers have plenty of amorphous domains to get elongated by strain before the small crystalline domains break apart as the amorphous domains are softer. Furthermore, the amorphous region of the non-conjugated polymers may be much lower in modulus and therefore can dissipate strain energy more effectively through chain alignment. On the other hand, the elongated/aligned polymer chains may further undergo strain-induced crystallization. On the contrary, strain-induced amorphization may occur for high crystallinity samples or samples with large crystalline domains and rigid polymer chains.

Stretchable semiconducting polymer is a unique class of material with low crystallinity compared to metallic materials, but the amorphous domains may also be rigid due to the rigid conjugated backbones and small aggregates formation from π - π interactions. It was found previously that when

additives were incorporated (e.g. semiconducting DPP polymers with DOP molecules: ref#25), strain-induced crystallization was observed in the semiconducting polymer films due to reduction of small aggregates formation and more freedom for the polymer chains in the amorphous domains to re-orient. Therefore, we believe both strain-induced crystallization and amorphization are possible and indeed we have observed both phenomenon in neat semiconducting polymers. Certainly, a deeper investigation is needed in future work.

Changes made:

Additional descriptions have been added in page 4 of the revised manuscript to indicate the difference between conventional semicrystalline polymers and semiconducting polymers:

...Note that strain-induced crystallization is a more common phenomenon for semicrystalline polymers. Strain-induced changes on crystallinity in polymer semiconductors only became of interest recently and has been less investigated. But we have observed such a phenomenon in several polymer semiconductors we investigated previously and in current systems. The observed overall change of $rDoC$ on PSC thin films may be from a competition between strain-induced crystallization and strain-induced amorphization depending on the specific thin film crystalline domain fraction, size, and distribution as well as polymer backbone rigidity and level of local aggregation in the amorphous domains. Due to the high rigidity of conjugated structures, a higher stress is asserted on the crystalline regions during strain compared to the more flexible non-conjugated polymers....

4. Re Q6: The authors incompletely address my concern about Tg. I agree that crystallinity and molecular mass affect elastomeric properties but to observe rubbery behavior at room temp requires Tg below room temp. Reviewer 2 Q7 echoes my concern. Perhaps a statement within the main text speculating that the thin-film Tg differs from the bulk Tg would reconcile the observations?

Response: We appreciate the comment from the reviewer. Tg for PSC thin film (<100 nm used for our studies) is difficult to measure. We have revised the description in page 6 of the revised manuscript to clarify that Tg measured here was from the bulk films (>1 μm): *...The glass transition temperatures (Tg) of the conjugated backbone were only visible for bulk P3 and P4 films (>1 μm in thickness, Supplementary Figure S2). This implies that P3 and P4 thin films are likely more disordered. Note that the Tg of sub-100 nm thin film (i.e. thin film for stretchable transistor device), which usually requires specialized methods to measure, are known to differ from the bulk film (Macromolecules, **47**, 3497-3501 (2014))*

Reviewer #3 (Remarks to the Author):

The authors have revised this paper taking into account all my suggestions. Furthermore, the paper was revised to enhance clarity of the presentation of significance to the field of thin-films for TFT/optoelectronics. I believe this work can be published without additional modifications.

Response: We appreciate the comment from the reviewer.

Reviewer #4 (Remarks to the Author):

The manuscript introduces a DPP-based highly stretchable polymer semiconductor (P4) that can be biaxially stretched up to 100% strain with stable electrical performance by dissipating the energy through multiple mechanisms. The authors develop a new metric, relative stretchability (rS), to quantitatively evaluate the potential for polymer semiconductors (PSC) in stretchable electronics, especially for thin film transistors. To prove that PSC with high rS has better and reversible performance, the authors reported a new polymer with a series molecular weights (P1-P4). I believe the P4 polymer has quite impressive performance and shows great potential for the stretchable transistors. Also, it is important to have a new metric to evaluate the PSC that can incorporate existing parameters, like elastic modulus, the crack on set strain (COS), crystallinity and others. However, there is not enough evidence to prove that rS can more accurately predict the performance of PSC. The idea is good, but I have questions and concerns about validation of this metric, specifically its generality. I would recommend this for publication if the concerns and the questions are properly addressed.

Q1. In page 5 line 112, reference 37, 40, and 41 are irrelevant to the T_g, if the references are mislabeled, please correct them.

Response: Thank you for the comment. The labels of the references have been corrected since these references are referred to the molecular weight and molecular ordering.

Q2. In the abstract and in page 8, the authors claim that “PSC with low rS value showed low crack onset strain” and “a larger rS value indicates more mechanical energy can be dissipated under strain through chain alignment or crystal alignment and corresponding a high crack onset strain (COS) and more reversible electrical performance of the polymer” from Figure 1e. The rS of P1-P4 has a good correlation with the COS, however, it is not the case for other polymers. This rS and COS correlation seems to work the best for the designed PSC but with different Mw, when the backbone and side chain are different, it loses the generality. For DPP-based polymer, P1 has much higher rS value but significant lower COS than PDPPTVT, not mention PDPPTVT-alkyl which has more modifications on the backbone. There are some limitations to compare PSC across the polymer libraries with this metric. It is necessary to prove generality for a new proposed metric. There are plenty of commercially available and lab reported materials for thin film transistors, but how this metrics work for these other materials is currently an unknown.

Response: The reviewer is correct that the rS of P1-P4 did correlate with their COS values. In this case, P1-P4 have the same polymer structures and different molecular weight. Furthermore, all film preparation and substrate used are similar across samples. However, this is not always the case when comparing to other PSCs in the literature. Traditionally, COS was used to evaluate the stretchability of the PSC thin films, but the correlation between COS and electrical properties, such as mobility in FET devices, under strain was not always consistent. In most of the cases, a thin semiconducting polymer was characterized on a stretchable substrate because: (1) the actual semiconducting polymer layer thickness is below 100 nm in the devices and (2) such a polymer active layer needs to be integrated with a stretchable substrate. The change of COS of the PSC thin film on supported stretchable substrates may also be affected by the thickness of the PSC, the modulus mismatch, adhesion, and additional interactions (e.g. hydrogen bonding) at the interface between PSC layer and supporting substrate (*Nature Nanotechnology* 17, 1265–1271 (2022)). Therefore, the measured COS reported in literature may vary depending on sample preparation and experimental method while the data for P1-P4 were all collected using similar conditions. This is also one of the motivations for introducing rS, in addition to the traditional parameters used to characterize mechanical properties, to

allow comparison of various types of polymer semiconductor thin films that are relevant to stretchable electronics prepared from different types of polymers and different processing and sample testing conditions.

Additionally, we thank the reviewer suggestion of using commercially available conjugated polymers to systematically measure rS for more materials. Since extensive characterizations of each polymers are needed and $rDOC$ measurements are dependent on available beamtime from advanced X-ray facilities, we hope to carry out such a study over time. However, we believe the dataset we currently have, based on eight different types of polymer backbones (with DPP-8TVT added during revision) with various molecular weights, different degrees of crystallinity and sample preparation conditions, provides a good initial validation of the utility of rS to compare various materials. We note the point that additional datasets will be helpful in identifying any limitations and refinement of rS in future work in the revised manuscript.

Changes made:

We have removed the “COS” term in the abstract and modified the description in page 8 of the revised manuscript as: *...In general, we found polymers with a higher rS tends to have improved electronic stability and mechanical durability under strain since more mechanical energy can be dissipated under strain through non-deleterious processes, such as chain alignment or crystal alignment (Figure 1e). Note that values of crack onset strain may be subjected to the sample preparation conditions, measuring conditions, and the calibration methods used. Moreover, rS values, in the range from 0.3~3, were calculated for other reported stretchable PSCs based on published data. Similarly, their rS values correlated well to the trend of electrical properties under strain as reported in the literature (Figures 1f and g).¹⁵⁻¹⁸ Therefore, rS provides a reasonable comparison of various designs of stretchable PSCs based on their morphological response to strain. Future work will be carried out to collect more additional datasets to identify any potential limitations and refinement of rS *

Q3. In Figure 1 f and g, the correlation of normalized mobility and rS seems good at low rS value. As there is no data point in the middle range (5-10) and P4 appears to be an outlier when compared to the other points. It would help to better describe the trend to have a few data points to cover the whole rS range if possible (from 0 to 10). Doing so would show that while P4 appears to be off trend, it is properly represented.

Response: Most of the reported stretchable neat PSCs showed a rS value between 0 to 5, as summarized in Figure 1d. A neat PSC with relevant morphological characterization data and a calculatable rS between 5 to 10 are rare in literature. However, a terpolymer, DPP-8TVT, in ref 19 showed a rS of 7.1, as depicted in Figure 1h. During this revision, we have extracted the mobilities of DPP-8TVT under 50 and 100% strain and compared them with the results of neat PSCs in Figures 1f and g, as shown in Figure S10 in the revised Supplementary Information. The DPP-8TVT results follow the general trend of neat PSCs in terms of normalized mobility under strain and rS values, suggesting the results of P4 is not an outlier. As a future work beyond this study, we agree more neat PSCs with a rS value of 5 to 10 need to be explored as the reviewer suggested.

Changes made:

The comparison between DPP-8TVT and reported neat PSCs in terms of rS and normalized mobility under strain has been added in the revised Supplementary Information as Figure S10, and additional descriptions have been added in page 9 of the revised manuscript:

...Additionally, the normalized mobility under strain of DPP-8TVT FETs with a rS value between 5-10 also followed the general trend of other neat PSCs based on their rS values (Supplementary Figure S10), suggesting the results of P4 is not an outlier....

Figure S10 | Correlations between charge carrier mobility and rS values. Normalized FET mobility under (a) 50 and (b) 100% uniaxial strain (i.e. mobility under 50 or 100% strain divided by the mobility at 0% strain) of the studied and reported PSCs compare with terpolymer DPP-8TVT.

Q4. It is unclear whether the mobility plotted in Figure 1 f and g is parallel or perpendicular to the strain direction or the average of both. From Figure S11, P1-P4 seems to have isotropic charge transportation under strain. Some polymers have anisotropic charge transport under uniaxial strain. How does the mobility vs rS plot of those polymers look in both directions?

Response: The mobilities in Figures 1f and g are calculated by averaging both the parallel and perpendicular directions since the rDoC we used for rS calculation extracted the molecular ordering from both parallel and perpendicular to the strain direction. It will be interesting to differentiate the mobility to parallel and perpendicular directions as a follow up study in the future because the rS will also need to be re-calculated based on the molecular ordering respect to different strain directions.

Q5. The authors state that rS shows good correlation with the reversibility of electrical performance as well when rS was first introduced in the main text, page 7. The only reversibility I can tell is rDoC, the molecule ordering. However, there is no direct evidence which shows that reversibility of the electrical properties and mechanical properties were related to the rS . As all electrical properties (mobility) plots are under strain, when the authors want to claim the reversibility, it will be helpful to have the plot of mobility vs rS at the strain release condition and/or some physical supports.

Response: We thank the reviewer for this comment. Indeed, P4 with the highest rS showed reversibility in mobility after 500 stretch/release cycles. The comparison on electrical properties across PSCs in different works is complex because a field-effect transistor (FET) contains multiple layers including substrate, gate electrode, dielectric layer, active charge transport layer of semiconducting polymer, and source and drain contacts. The thickness of semiconducting polymer

layer is typically 30-100 nm since charge transport is mostly confined in the first 5 nm-thick of film near the semiconductor-dielectric interface. The charge carrier mobility can be easily affected by the device fabrication and processing methods, especially under the strain-released condition. Alternatively, rDoC is a factor that is highly related to the mobility because the molecular ordering generally possesses positive correlation to the PSC charge transporting. Indeed, the reversibility of crystallinity in Supplementary Figure S31 showed a good indication that a PSC with high rS value exhibited good reversibility in electrical properties. As seen from Supplementary Figure S27, mobility of P4 can be maintained after 500 stretch/release cycles under 100% biaxial strain, which has not yet been achieved by other PSCs.

Q6. A follow-up question for reversibility in mechanical properties. P2, P3 and P4 all have clear yield points from 2% to 5% and P4 has a highest yield stress. With such a clear yield point and a high yield stress, the statement of the mechanical reversibility is surprising. Although the mechanical behavior can change from bulk to thin film, a full cycle of stress-strain curve (before the elongation at break) would back up the statement. At least the curve for P4 polymer should be provided. It will be more necessary to provide the stress-strain curve of the thin film if the authors want to make a point on the mechanical properties of thin film. In addition, no other morphology results except the GIXD results of the strain released films were provided. If the thin film is transferred to a PDMS substrate, a simple microscope image of strain released film should be accessible.

Response: We thank the reviewer for this comment. Indeed, the mechanical properties of bulk and thin semiconductor films would be different. The tensile test samples were prepared by drop-casting on top of a Teflon plate with a thickness of 2-3 μm . With such a thick polymer film, the focus of the test was to demonstrate our PSCs can be stretched even as a bulk film and to correlate the trend of the elongation at break values to their molecular weight as well as degree of polymer entanglements. The trend should persist in thin films, even though the absolute values of elongation at break is quite different from those measured for thin films. The thin PSC films for stretchable electronic devices with a thickness of approximately 50 nm are difficult to be performed for conventional tensile test as they are too thin and fragile for the sample mounting or clamping set-ups. Therefore, thin PSC films in the study were stretched on the PDMS support substrate, and the stress-strain curve of the PSC thin film could not be extracted from a PSC thin film/PDMS sample due to the big difference of thickness between PSC thin film (50 nm) and PDMS substrate (0.5-1 mm). Stress-strain curves for a P4 bulk film under cycles of various strains are provided in Figure R2 as a reference, hysteresis is observed since the bulk P4 film yielded below 5% of strain. A method to extract the stress-strain property of PSC thin film will be important to develop as a follow-up work in the future.

Figure R2 | Stress-strain curves for a P4 bulk film (thickness of 2-3 μm) under cycles of various strains measured using Instron.

As suggested by the reviewer, optical microscopy images of strain-released P1-P4 films have been added in the revised Supplementary Information as Figure S7. Damages can be observed on lower molecular weight and low rS value P1 and P2 films, while P3 and P4 films stayed smooth and continuous without noticeable defects.

Figure S7 | Optical microscopy images of strain-released P1-P4 films. The thin films (approximately 50 nm) supported on PDMS substrates were strained to 100% strain and then released for these microscopic images. Damages can be observed on lower molecular weight and low rS value P1 and P2 films, while P3 and P4 films stayed smooth and continuous without noticeable defects.

Q7. In page 11, the authors mentioned that the edge on packing with strain is increasing monotonically for P3 and P4, however, in Figure 3d the packing orientations suggests that P3 has monotonic increase in the edge on packing and the edge on packing of P4 appears to plateau from 25% strain with a fraction of 20%. From my point of view, the energy may not be dissipated through the crystallites rotation since 25%. This behavior does not exactly match the statement made in the context. The authors should be cautious when providing an explanation for this observation.

Response: We appreciate the reviewer's comment. The mechanical energy can be dissipated by multiple pathways under strain, such as (i) elastic or plastic deformation of the amorphous domains, (ii) molecular alignment or re-orientation of crystallites, (iii) amorphization of crystalline domains, and (iv) ultimately bond breakage and crack formation. As the reviewer mentioned, the change of packing orientation of P4 under strain was not substantial when the strain was larger than 25%, which is probably because the DoC of P4 is lower than P3 (Figure S5). With a lower DoC, the deformation of the amorphous domains may contribute more to the overall energy dissipation (i.e. slightly higher change of degree of aggregation under strain was observed (Figure S10)), leading to less changes on the packing orientation under strain.

Changes made:

We have modified the description in page 11 of the revised manuscript to take into account comments from the review: *...Similarly, a monotonic increase in edge-on packing with strain was observed for polymers with higher molecular weight, especially with P3 film. Without microcracking (Supplementary Figure S6b), the face-on and edge-on ratio continued to change and contribute to energy dissipation up to 100% strain. Note that P4 film exhibited lower level of changes on the packing orientation probably due to its low DoC (Supplementary Figure S5)....*

Q8. In Figure 2b, the rDoC of P2 keeps dropping from 60% to 50% when the 100% uniaxial strain is released, but in Figure S21, the rDoC of P2 recovers from 20% to almost 50% when the 100% biaxial strain is released. If the rDoC increased after the strain is released, does that mean the crystalline domain in P2 is recovered under biaxial strain but not under uniaxial strain? In addition, the coherence length of P2 in perpendicular direction under uniaxial strain is pretty much the same as the coherence length in biaxial strain. The authors used the orthogonal compression to explain the drop in coherence length at 100%, but when the film is uniformly biaxially stretched, what breaks the domain? These behaviors are different from what I would have expected, and I hope the authors can clarify these observation.

Response: The decrease of rDoC under strain can be observed when the crystallites are amorphized or the cracks damage the ordered domains. When the PSC film is released under a uniaxial strain, the counter force of orthogonal compression can further act on the ordered domains so that the rDoC was further reduced. However, the stretching forces were isotropic applied to the PSC film under biaxial strain, the counter forces during the film releasing will also be distributed biaxially on the film. Thus, it is possible that some of the amorphized crystallites were re-crystallized as the strain released and reflected to a recovery of rDoC since the forces were not applied to a particular direction.

Changes made:

We have modified the description in page 13 of the revised manuscript to clarify the decrease of rDoC and coherence length can be due to the crystalline domains in the film were amorphized or cracked: *...Similar to the results of uniaxially stretched films, lower molecular weight P2 films showed*

substantial decrease in rDoC and coherence length under biaxial strain, indicating the crystalline domains in the film were mostly amorphized or cracked....

On the other hand, there may still be some compression forces in the biaxially strained film from the side of the PSC film under biaxial stretching since the PSC film was stretched from its four corners (Supplementary Figure S21). Indeed, the uniaxial and biaxial PSC films were deformed with the same mechanism. The only difference is the cracks were formed perpendicularly to the strain direction under uniaxial stretching since the stretching forces were applied to a distinct direction (Supplementary Figure S6); while the cracks on biaxially-stretched film were seen as isotropically oriented (Supplementary Figure S22).

REVIEWERS' COMMENTS

Reviewer #1 (Remarks to the Author):

I thank the authors for their consideration and adoption of my suggestions. I am satisfied that my concerns have been addressed and I recommend the manuscript proceed to publication.

Reviewer #4 (Remarks to the Author):

All of my comments are properly addressed and the manuscript was revised accordingly. I think this manuscript is fit for publication. In the future, a further work that can expand and/or refine rS would be beneficial.

We appreciate the feedback from the reviewers. Below is a point-to-point response to their comments:

Reviewer #1 (Remarks to the Author):

I thank the authors for their consideration and adoption of my suggestions. I am satisfied that my concerns have been addressed and I recommend the manuscript proceed to publication.

Response: Thank you very much for the comment from the reviewer.

Reviewer #4 (Remarks to the Author):

All of my comments are properly addressed and the manuscript was revised accordingly. I think this manuscript is fit for publication. In the future, a further work that can expand and/or refine rS would be beneficial.

Response: Thank you very much for the comments and suggestions from the reviewer.